# Localized Convolutional Neural Networks for Geospatial Wind Forecasting

**Arnas Uselis \*, Mantas Lukoševičius \*** and **Lukas Stasytis**

Faculty of Informatics, Kaunas University of Technology, LT-51368 Kaunas, Lithuania
\* Correspondence: auselis@gmx.com (A.U.); mantas.lukosevicius@ktu.edu (M.L.)

**Abstract:** Convolutional Neural Networks (CNN) possess many positive qualities when it comes to spatial raster data. Translation invariance enables CNNs to detect features regardless of their position in the scene. However, in some domains, like geospatial, not all locations are exactly equal. In this work, we propose localized convolutional neural networks that enable convolutional architectures to learn local features in addition to the global ones. We investigate their instantiations in the form of learnable inputs, local weights, and a more general form. They can be added to any convolutional layers, easily end-to-end trained, introduce minimal additional complexity, and let CNNs retain most of their benefits to the extent that they are needed. In this work we address spatio-temporal prediction: test the effectiveness of our methods on a synthetic benchmark dataset and tackle three real-world wind prediction datasets. For one of them, we propose a method to spatially order the unordered data. We compare the recent state-of-the-art spatio-temporal prediction models on the same data. Models that use convolutional layers can be and are extended with our localizations. In all these cases our extensions improve the results, and thus often the state-of-the-art. We share all the code at a public repository.

**Keywords:** convolutional neural networks; recurrent neural networks; deep learning; machine learning; spatial-temporal wind forecasting

---

## 1. Introduction

To help fight climate change, energy production increasingly turns to clean renewable energy sources like solar and wind. A fundamental constraint of these sources is, however, that their energy production is stochastic and directly depends on the ever-changing weather conditions. Wind can be especially volatile. Consequently, matching the energy demand and supply in the grid at every time is getting more challenging. This is being addressed by emerging smart grid technologies where nodes can produce, consume or store energy, communicate with other nodes, and the energy is being market-priced in real time. The ability to forecast wind or solar energy production is an essential part of this solution. For wind, this boils down to forecasting wind speeds at the locations of the wind turbines.

In addition to classical Numerical Weather Prediction (NWP) models, machine learning approaches are increasingly applied, especially for particular locations and short time spans ("nowcasting") where the longer-term NWP models are not available or are expensive to update and rerun. The wind prediction at multiple locations (often spatial grids) given the historical data and possibly other types of signals, is a spatio-temporal (or geo-temporal) task, as it involves both spatial and temporal components. Many deep learning architectures have been recently proposed for this type of task [1].

When predicting a regular grid, the spatial component of the task is typically handled by the convolutional layers in the deep architectures, the eponymous building blocks of Convolutional Neural



Networks (CNNs) [2]. Convolutional layers have a very nice property, that they treat each location equally and learn, share the same weights at each. This is very helpful, as the laws of atmospheric physics are the same at each location, but only to a point, as locations may also have their special intrinsic features.

In this contribution, we propose to enhance convolutional layers with several flavors of localized learnable features to strike a balance and benefit from both location-invariant and location-specific learning. We test our proposed approaches first on a synthetic benchmark dataset and then on three wind forecasting datasets rigorously comparing to the state-of-the-art. Our localized convolutions consistently increase the performance of the corresponding non-localized architectures and improve the state-of-the-art.

We share all the code at a public repository https://github.com/oshapio/Localized-CNNs-for-Geospatial-Wind-Forecasting (Supplementary Materials).

This article is organized in the following way. We provide our motivation and conceptual idea of the localized CNNs in Section 2. We systematically review related previous work in Section 3, both state-of-the-art geo-temporal prediction architectures organized by the way they integrate the spatial and temporal aspects (Section 3.1) and previous attempts at localizing CNNs (Section 3.2). We introduce our proposed methods of localizing CNNs in Section 4, moving from more specific to more general implementations. We provide detailed descriptions of the state-of-the-art deep neural network architectures that we use in our numerical experiments, together with our localized CNN modifications in Section 5. One synthetic and three real-world datasets, their experiment specifics, and the results with the different models are presented in subsections of Section 6. There we also propose a mutual-information-optimized embedding of the data that were originally not on a regular grid in Section 6.3.1. The article is concluded with the discussion in Section 7.

## 2. Localized CNN Motivation

Convolutional Neural Networks (CNNs) [3] are deep neural networks applied to spatially-ordered data, like images. They have become a staple of deep learning and image processing. Their defining component is the eponymous convolutional layer where each neuron has a fixed local receptive field of the layer input and shares its weights with all the other (repeated) neurons arranged in a lattice corresponding to the dimensions of the input. This group of identical neurons can alternatively be seen as a local filter (or "kernel") having the size of the receptive field that is convolved with the input in 2D to compute the activations. A convolutional layer typically consists of multiple such filters. More specifically, convolution in 2D is defined as

$$(f * g)(i, j) = \sum_{m=-\frac{k}{2}}^{\frac{k}{2}} \sum_{n=-\frac{k}{2}}^{\frac{k}{2}} f(m, n) g(i + m, j + n), \tag{1}$$

where $f(\cdot, \cdot)$ is some $k \times k$ size filter applied to every $k \times k$ size patch centered at $(i, j)$ on the input image $g(\cdot, \cdot)$. Typically, an element-wise nonlinear function is applied to the results of the convolution which gives a lattice of identical weighted-sum-and-nonlinearity neurons each looking at a different $k \times k$ size patch of the input image.

Convolutions give CNNs some unique benefits, compared to regular MultiLayer Perceptrons (MLPs):

- The total amount of different trainable weights is drastically reduced, thus the model is:

    - Much less prone to overfitting;
    - Much smaller to store and often faster to train and run;

- Every filter is trained on every $k \times k$ patch of every input image $g(\cdot, \cdot)$, which utilizes the training data well;

- The architecture and learned weights of the convolutional layer do not depend on the size of the input image, making them easier to reuse;
- Convolutions give **translation invariance**: the features are detected the same way, no matter where they are in the image.

Translation invariance (or more precisely equivariance, when talking about a single convolutional layer) is very important for good generalization when objects being detected are randomly framed in the images, which is common, as it allows us to detect each object or feature equally well regardless of location in the frame.

Translation invariance, however, is absolute in convolutional layers: it is not that they can learn to treat each location the same, but they absolutely must do so and are structurally unable to learn to do otherwise. There is simply no mechanism for that in (1): the filter $f$ has no way to "know" the location $(i, j)$ it is being applied to, nor a way to learn and store some location-specific information. Thus, the benefit can also become a limitation.

The complete translation invariance (or "location agnosticism") might not always be optimal even for images as not all objects or features are equally likely to appear in every part of them. This is especially evident when the images have a constant static framing, like passport or mugshot photos of centered faces (even more so if they are specifically aligned [4]), or frames from a stationary security or road camera. Statistically, this is also true for photo or video images in general: for example, eyes are likely to appear above a mouth and a nose, or sky is more likely to appear in the upper parts of the image than the lower. Models with an appropriately learned bias could result in a better recognition or a more frugal implementation.

As a very particular example, a convolutional layer would not be able to learn to ignore a dead pixel in a camera or a static watermark, TV channel logo at a specific location. It would not be able to exploit the fact that the artifact is always located at the same place in all the pictures, but would treat it as a feature that could appear anywhere in the frame.

There are more applications where the convolutional neurons would benefit from "knowing where they are" on the lattice/image, as pointed out in [5].

The complete translation invariance is also usually sub-optimal in geospatial CNN applications, like meteorological forecasting, where the locations of the lattice points are fixed. While the same laws of atmospheric physics apply at every location, each location typically also has its own unique features like altitude, terrain, land use, large objects, sun absorption, the heat capacity of the ground, heat sources, etc. that affect the dynamics. This becomes even more relevant, when the covered area increases, and so do the differences and variety of the locations, including local climate factors.

In general, similar dynamics or patterns, features could manifest themselves (slightly) differently at different locations of the input lattice, and we need an adequate "localized", not completely location-agnostic CNN model that could learn to account for this.

The complete location agnosticism in CNNs can be remedied in several ways by supplying different additional static location-dependent features:

1.  The explicit coordinates of each location like in CoordConv [5];
2.  A combination of local random static location-dependent inputs, that could potentially allow us to uniquely "identify" each location as well;
3.  The above mentioned real-world relevant unique location-specific features if they are explicitly available (typically not all of them are).

Options 1 and 2 only give a chance for the convolutional neurons to "orient themselves" on the lattice, but not much more. Option 3 provides some local features relevant to the task and should probably be used in any case whenever they are available. The convolutional neurons have to learn how to incorporate all this information in a meaningful way.

We could, however (or in addition), allow the model to learn the location-based differences more directly by introducing additional:

1.　Learnable local inputs/latent variables,
2.　Learnable local transformations of the inputs in the form of local weights at every input lattice location.

These enhancements would allow the model to learn some useful features or tendencies of each location and store them in a form of learned parameters locally. They enable us to have "Localized CNNs": a model that learns to treat the different locations/pixels in the input similarly, but not identically. This is the main theoretical/methodological contribution of this article, which is described in detail in Section 4.

There may be other ways of producing localized CNNs in addition to the ones described above.

We do not aim to eliminate translation invariance and other mentioned benefits of CNNs completely, which could easily be done by reverting back to regular MLPs or locally connected layers, but to strike a balance by making translation invariance non-absolute and retaining the other useful features of CNNs to the extent possible or needed. We enable this, but leave the extent it is used partially up to the training algorithms.

## 3. Related Work

In this section, we review related previous work. It is split into three parts. In Section 3.1 we provide the formalism to build on and a systematic review of state-of-the-art architectures that combine spatial and temporal aspects of the predictions in this domain. The spatial components of the architectures involve convolutional layers where our localizations are applicable. In Section 3.2 we review the few past efforts of localizing CNNs that are most relevant to our approaches and in Section 3.3 mention several previous cases on input learning in deep networks.

### 3.1. Geo-Temporal Prediction

In this research, we tackle wind speed prediction at many stations, given their history. For instance, these stations could be sensor data collected from wind turbines, where the forecasting at each site is especially important [6].

We want to forecast wind speeds at a given regular rectangular spatial grid with dimensions $W \times H$, where $W$ and $H$ are the width and the height (or rather length) of the grid, respectively. We assume that the observations are made with a uniform sampling rate in time, and there are $T$ time steps in total. This problem is referred to as spatio-temporal forecasting on a regular grid [1]. Here we only consider the wind speed, so the observations at the time instance $t$ can be expressed as a matrix $\mathbf{X}_t \in \mathbb{R}^{W \times H}, t \leq T, t \in \mathbb{N}$. At the time step $t$ we are interested in predicting $\mathbf{X}_{t+h} \in \mathbb{R}^{W \times H}$, where $h$ is the forecasting horizon, given an $l$-window of observations from the past. This can be expressed as

$$\tilde{\mathbf{X}}_{t+h} = \underset{\hat{\mathbf{X}}_{t+h}}{\arg\max}\, p(\hat{\mathbf{X}}_{t+h}|\mathbf{X}_{t-l+1}, \mathbf{X}_{t-l+2}, ..., \mathbf{X}_t), \tag{2}$$

where $p(\cdot|\cdot)$ denotes the probability of a particular state, given the history. Multi-step forecasting could be defined in a similar way, as seen in [7].

From an application perspective, there is a vast body of research focused on spatio-temporal problems. For instance, spatio-temporal relations are modeled in traffic forecasting [8,9], precipitation nowcasting [7], air-quality prediction [10] and other fields.

Many of the applications have been inspired by the progress in video prediction and classification, since raster spatio-temporal data can be viewed as a video with an arbitrary number of color channels (in our case a single one). For instance, one of the first large-scale applications of CNNs in video classification was in [11], while the combination of CNNs and Recurrent Neural Networks (RNNs) has been introduced in [12].

In most of the contemporary research for wind characteristics forecasting, the spatio-temporal problem is usually solved by a connected spatial and temporal components of the architecture. A time

window of the input frames are usually fed to the spatial component, and the representations produced by it are then fed to the temporal component. This can be interpreted as a spatial encoder and a temporal decoder, especially when several prediction horizons are simultaneously produced.

One way to implement this is to apply an identical spatial model to each frame and then feed a time window of the resulting representation to a temporal model, that combines these features in a meaningful way over time and produces the prediction matrix:

$$\tilde{\mathbf{X}}_{t+h} = f_{\text{temporal}}\left(f_{\text{spatial}}(\mathbf{X}_{t-l+1}), f_{\text{spatial}}(\mathbf{X}_{t-l+2}), \dots, f_{\text{spatial}}(\mathbf{X}_t)\right), \quad (3)$$

where $f_{\text{spatial}}(\cdot)$ is the model extracting spatial features, usually the standard CNN, and $f_{\text{temporal}}(\cdot)$ is the temporal model with spatial features as input, typically, an RNN model or a simpler memory-less feed-forward neural network, where the representations from the spatial encoder are concatenated. Here the spatial encoding is not time-specific, which is probably adequate, considering that we use a sliding temporal window in input and a capable temporal model above this layer.

This approach has been used in a model named Predictive Deep Convolutional Neural Network (PDCNN) [13] where a CNN followed by an MLP was applied to forecast wind speeds in a farm of 100 wind turbines aligned in a rectangular grid and demonstrated superior results to classical machine learning techniques. Shortly after, the researchers introduced a follow-up model consisting of a CNN followed by a Long Short-Term Memory (LSTM) [14] RNN, called Predictive Spatio-Temporal Network (PSTN) [15], demonstrating increased performance compared to both the PDCNN and the LSTM alone.

Alternatively, time can be taken as an additional dimension of the grid and a spatial encoding done of the entire time window:

$$\tilde{\mathbf{X}}_{t+h} = f_{\text{predictor}}\left(f_{\text{spatial}}([\mathbf{X}_{t-l+1}, \mathbf{X}_{t-l+2}, \dots, \mathbf{X}_t])\right), \quad (4)$$

where $[\cdot, \cdot, \dots, \cdot]$ denotes concatenation, $f_{\text{spatial}}(\cdot)$ is a model that treats the concatenated over time inputs as spatial but having $l$ times more input channels, typically a classical CNN, and $f_{\text{predictor}}(\cdot)$ is a model interpreting the spatial representation, usually a feed-forward neural network, to make the predictions. This approach empowers the spatial component to combine inputs along time. Note, that $f_{\text{spatial}}$ here does no convolution over the time dimension, but instead all its filters see all the time steps. Thus it produces and passes to $f_{\text{predictor}}$ a representation with no temporal dimension. Conversely, if every filter in $f_{\text{spatial}}$ would only see a single time step, (4) would fall back to (3).

Not all the architectures used for this type of task have both the spatial and the temporal component. For example, the $f_{\text{predictor}}$ in (4), which is a somewhat degenerate case of $f_{\text{temporal}}$ in (3), can be dropped resulting in a fully convolutional model

$$\tilde{\mathbf{X}}_{t+h} = f_{\text{spatial}}([\mathbf{X}_{t-l+1}, \mathbf{X}_{t-l+2}, \dots, \mathbf{X}_t]). \quad (5)$$

The authors of [16] have applied the DenseNet [17] architecture as $f_{\text{spatial}}$ to predict wind power. The data were non-spatially-ordered, but to use it with CNNs, they embedded the wind turbines into a rectangular grid based on their relative positions. The authors examine two model variations: FC-CNN, which is a model of type (3) with $f_{\text{predictor}}$ being a fully-connected layer, and E2E, which remains fully convolutional as in (5).

On the other hand, the problem can be treated as a purely temporal one, ignoring the spatial nature of the data

$$\tilde{\mathbf{X}}_{t+h} = f_{\text{temporal}}([\mathbf{X}_{t-l+1}, \mathbf{X}_{t-l+2}, \dots, \mathbf{X}_t]), \quad (6)$$

by using, e.g., a purely RNN or MLP model.

Combining the spatial and temporal components in an architecture is not a trivial task. In the approaches discussed so far, even though the spatial component (CNN) takes into account the spatial

information, the usual choices of the temporal component (MLP or RNN) do not. This seems somewhat sub-optimal, especially here, since the output of the whole model is still spatial, just as its inputs.

This problem has been addressed first by introducing Convolutional LSTM (ConvLSTM) in [7], whereas not only the CNN but also the LSTM support spatial-order-preserving operations. To achieve this, the temporal portion of the LSTM's state is a 2D matrix whose entries correspond to the topology of the spatial input. This architecture has been applied successfully to wind power forecasting [18].

A slightly different approach has been taken in [19] where the authors had non-uniformly embedded spatial data and, instead of explicitly embedding it onto a grid and making use of the CNN to model spatial relations, used a CNN to extract the feature relations between multiple weather factors.

### 3.2. Previous CNN Localizations

Several recent contributions propose appending the coordinates to the spatial representations to aid the CNN localization. A CNN with two additional input channels indicating the x and y coordinates of each input in the 2D plane, called CoordConv, was proposed in [5]. The authors demonstrated that this substantially improved the performance compared to standard CNN when tasks involve a need for localization.

Similarly, adding location coordinates to the CNN representations, called a semi-convolutional operator, was shown to help separate different instances of the same type of object in an image pixel-wise segmentation task in [20]. A spatial broadcast decoder has been successfully proposed as the decoder part of a deep variational autoencoder, where it tiles the encoding with the appended x and y coordinates on a 2D lattice and uses a CNN instead of the usual deconvolutions in [21].

These approaches of adding additional coordinates belong to the list of static CNN localizations given in Section 2. We propose an alternative static localization of adding random input maps. While we add these static localizations to our experiments for comparison, our main contribution and interest are in learnable CNN localizations.

Similar CNN localization ideas to our learnable local transformations motivated in Section 2 and detailed in Section 4 have been successfully applied to face recognition in [4]. The authors align the faces before the recognition and use locally connected neural layers [22] after the CNN layers in their architecture to aid the recognition of specific parts of the faces. Contrary to the geo-temporal applications, the faces are not pixel-perfectly aligned, thus localizations only in the higher levels of CNN make sense. However, contrary to our methods, in that approach the convolutions and localizations do not mix, the latter follow the former.

### 3.3. Deep Input Learning

Input or latent variable learning in deep networks is not that uncommon and is often related to transfer learning.

There are some examples of specialization with deep MLP neural networks for language recognition by learning "conditional codes" as an additional input unique to every speaker in a mixed MLP/Hidden Markov Model [23], mixed MLP/CNN model [24], or as direct inputs to all the layers of an MLP [25]. The authors note, however, that this method does not work well with adapting CNNs.

Input optimizations of CNN have also been explored in neural style transfer [26] or deep dream (https://github.com/google/deepdream) applications, but this is done for a different purpose and it is not an additional input.

Meanwhile, learning additional/intermediate inputs to (a part of) a model that are transformations of the external inputs from data is abundant in deep leaning, and, in fact, is the fundamental principle of it.

Contrary to our learnable inputs approach, these deep input learning examples were used for different purposes than localization. The above mentioned "conditional code" personalizations are perhaps the closest in spirit, but are not applied to locations or CNNs.

## 4. Proposed Methods

We propose a few simple building blocks compatible with any kind of CNN architecture to help it better capture location-specific trends in the data.

### 4.1. Learnable Inputs

As mentioned before, CNNs by default are not able to identify the absolute position of the kernel. One way to amend that it to introduce **Learnable Inputs** (LIs) of the same spatial dimension that can be concatenated to the original inputs going to a convolutional layer. These static LIs are free parameters that themselves can be trained by backpropagation together with the rest of the network weights and do not require any kind of prior knowledge of the task.

More precisely, let the input array have a spatial dimension of $w \times h \times n$, where $w$ is the width, $h$ is the height and $c$ is the number of features of the array. We add $c$ LIs of the dimension of $w \times h$ each. Next we concatenate the input array with the learned map and yield a resulting array shaped $w \times h \times (c + n)$. This is depicted in Figure 1.

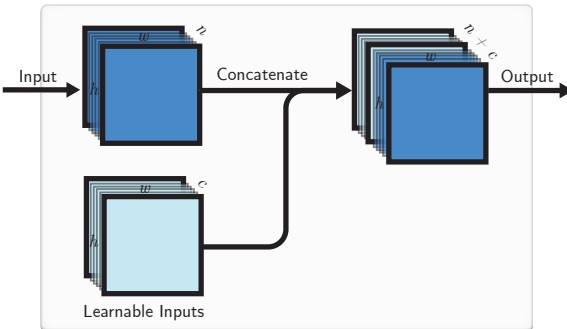

**Figure 1.** The proposed Learnable Input (LI) block. Each LI has the same spatial dimension as the original input. They are concatenated to the external input and passed to the other layers of CNN.

In the training process, LIs evolve simultaneously with the weights of the model and likewise are fixed after the learning phase, and the fixed learned maps are then used in other phases. The benefit of this is that now LIs can correspond to some local information extracted from training data about every location of the input and so more powerful models can be trained. For instance, Ref. [5] have shown that adding coordinates as an additional input helps the CNN to generalize at certain tasks where localization is needed, and if this is not needed, the kernels responsible for interpreting the coordinate information learns to ignore them. We note that these LIs can learn such a representation that would help the CNN to navigate better.

The $c$ input-sized LIs add extra $w \cdot h \cdot c$ learnable parameters to the model, plus the additional corresponding input weights of the next layer.

One way to implement LIs in modern deep learning frameworks is to add a constant unitary input and connect it with learnable local weights to get the LIs.

### 4.2. Local Weights

LIs might be not enough to enable CNNs to treat inputs at different locations differently. For this, we introduce **Local Weights** (LWs) as a locally connected layer of weights [22] that are not shared like in convolution.

One particular interpretation of LWs could be as importance maps that weigh locations of the input that are always more important to the task higher and others lower. This could be useful in cases where not all the regions of the input image are statistically equally important, as discussed previously. In an extreme case of this LWs could become input masks. In less extreme cases they could correct

for location-specific biases. However, in general, LWs allow for learnable local transformations of the input when similar features manifest themselves (slightly) differently in different locations.

LWs are learned in the training phase jointly with the rest of the network weights in the same way as LIs. In our proposed layer we concatenate the original inputs to the locally-weighted ones before passing them to any kind of further CNN layers (Figure 2), but this is not necessary.

More precisely, we define a multiplication operation $\otimes_{(1,1)}$ for the incoming input $X \in \mathbb{R}^{w \times h \times n}$ and local weights $M \in \mathbb{R}^{w \times h \times n \times d}$, which produces locally-weighted input $I \in \mathbb{R}^{w \times h \times d}$

$$I_{ijo} = f(X \otimes_{(1,1)} M)_{ijo} = f\left( \sum_{k=0}^{n} X_{ijk} \cdot M_{ijko} \right), \tag{7}$$

and more generally $\otimes_{(A,B)}$ is defined on input weights matrix (tensor) $M \in \mathbb{R}^{w \times h \times n \times A \times B \times d}$ as

$$I_{ijo} = f(X \otimes_{(A,B)} M)_{ijo} = f\left( \sum_{k=0}^{n} \sum_{l=0}^{A} \sum_{m=0}^{B} X_{i+l,j+m,k} \cdot M_{ijklmo} \right), \tag{8}$$

where $i, j, o$ are row, column, and depth of the resulting matrix $I$ respectively, and $f(\cdot)$ is the element-wise activation function. In our experiments we used the identity function $f(x) = x$.

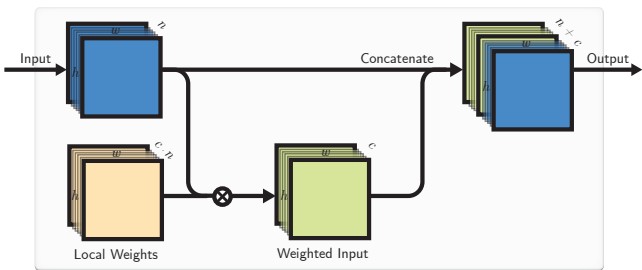

**Figure 2.** The proposed Local Weights (LW) block. Input (in blue) is multiplied (7) (depicted with $\otimes$) with the input weight block (in brown), and the weighted input is further passed to the other layers of CNN.

We make a distinction between local weights LWs and learnable inputs LIs. Since LIs are static, they cannot directly locally control the influence strength of the external inputs, and neither, more generally, the best local weighted combination of the external inputs, which LWs can. On the other hand, LIs convey special features of the location irrespective of the current inputs. Alternatively, LIs can be thought of as bias weights in LWs.

$d$ LWs with $1 \times 1 \times n$ receptive fields (7) add $w \cdot h \cdot d \cdot n$ learnable parameters to the model, which is $n$ times more compared to $d$ LIs.

In addition to the described LWs connected to $1 \times 1 \times n$ receptive fields of the input (7) and the more general, bigger $A \times B \times n$ ones (8), they could be even more direct element-wise weighting of the inputs with the receptive fields of $1 \times 1 \times 1$. We explore such variation by defining a $\odot$ operation on weight matrix (tensor) $M \in \mathbb{R}^{w \times h \times n}$ as a standard element-wise multiplication.

$$I_{ijo} = f(X \odot M)_{ijo} = f(X_{ijo} \cdot M_{ijo}). \tag{9}$$

*4.3. Combined Approach*

Finally, we investigate merging both the described CNN localizations LIs and LWs into a combined block illustrated in Figure 3.

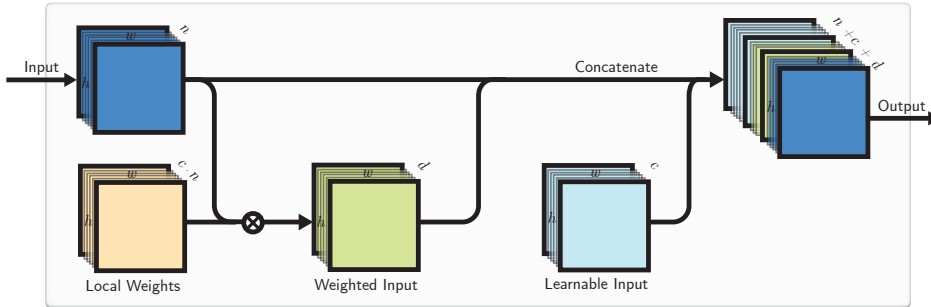

**Figure 3.** The combined approach.

This is, in essence, the LW block followed by the LI block. We can vary the number of LIs and LWs depending on the situation.

### 4.4. Implementation by a Locally Connected Layer

The combination of LIs and LWs can also be compactly implemented by a locally connected layer with biases, as illustrated in Figure 4. This layer is similar to the convolutional one, except that the weights here are not shared.

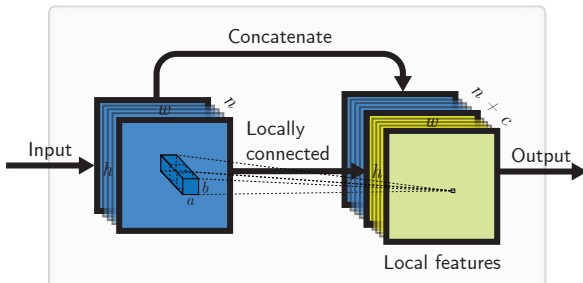

**Figure 4.** Implementation with a locally connected layer. Input goes through a locally connected layer with a $(a \times b)$ kernel and produces local features that are concatenated with the input.

The LIs here are somewhat enabled by the local bias weights. If the bias weights are not readily available, they can be implemented by appending constant unitary input channels to the input, before feeding them to the locally connected layer.

This approach, while it can potentially lead to a cleaner code in some current deep learning frameworks, somewhat sacrifices the fine control of the LIs and LWs that are meshed together here, and potentially have more (redundant) trainable weights for a similar effect.

## 5. Baseline and Localized Model Architectures

In this section, we introduce baseline models and their configurations that we use for comparison, together with our concrete localized CNN architectures. The objective is both to find the best model for a given dataset and also to show the compatibility and effectiveness of our proposed layers with these models. Note that we test these baselines only on real-world datasets.

The architectures that were explored are detailed in Table 1. We use a special notation for this. Functions denote layers here, and arrows denote the tensors passed between them with indication of their dimensions above. For more complex architectures, we define the parts of the model before defining the final model using them.

**Table 1.** Configurations of models used in experiments on real-world data. The localized models follow after the double line.

| Model Name | Architecture |
| --- | --- |
| Persistence **PR** | $Last(I)$. |
| **CNN** | $I \xrightarrow{W \times H \times T} C(5 \times 5 \times 45) \xrightarrow{W \times H \times 30} C(4 \times 4 \times 30) \xrightarrow{W \times H \times 30} C(3 \times 3 \times 30)$ $\xrightarrow{W \times H \times 30} C(1 \times 1 \times 1)$. |
| MultiLayer Perceptron **MLP** | $F(I) \xrightarrow{W \cdot H \cdot T} \sigma(FC(500)) \xrightarrow{500} FC(W \cdot H)$. |
| **LSTM** (adopted [13]) | $F(I) \xrightarrow{W \cdot H \times T} LSTM(300) \xrightarrow{300 \times T} Last(LSTM(W \cdot H))$. |
| **ConvLSTM** [7] | $I \xrightarrow{W \times H \times T} Last(ConvLSTM(50)) \xrightarrow{W \times H \times 50} C(1 \times 1 \times 1)$. |
| **PSTN** [13] | $C_t := F(I_t) \xrightarrow{W \times H \times 1} C(3 \times 3 \times 20) \xrightarrow{W \times H \times 20} Pool_{max}(2 \times 2) \xrightarrow{\frac{W}{2} \times \frac{H}{2} \times 20} C(3 \times 3 \times 50))$ $\xrightarrow{\frac{W}{2} \times \frac{H}{2} \times 50} C(2 \times 2 \times 200) \xrightarrow{\frac{W}{2} \times \frac{H}{2} \times 200} \sigma(FC(200));$ $[C_0, .., C_{T-1}] \xrightarrow{200 \times T} LSTM(300) \xrightarrow{300 \times T} Last(LSTM(W \cdot H))$. |
| **PDCNN** [15] | $C_t := F(I_t) \xrightarrow{W \times H \times 1} C(3 \times 3 \times 10) \xrightarrow{W \times H \times 10} Pool_{max}(2 \times 2) \xrightarrow{\frac{W}{2} \times \frac{H}{2} \times 10} C(4 \times 4 \times 30)$ $\xrightarrow{\frac{W}{2} \times \frac{H}{2} \times 200} ReLU(FC(30)); \quad [C_0, .., C_{T-1}] \xrightarrow{30 \times T} ReLU(FC(200)) \xrightarrow{200} \sigma(FC(W \cdot H))$. |
| **FC-CNN** [16] | $R_1 := I \xrightarrow{W \times H \times T} C(3 \times 3 \times 16); \quad B_1 := BN(R_1) \xrightarrow{W \times H \times 16} C(3 \times 3 \times 5);$ $R_2 := BN([R_1, B_1]) \xrightarrow{W \times H \times 21} C(3 \times 3 \times 16) \xrightarrow{W \times H \times 16} Pool_{avg}(2 \times 2);$ $B_2 := BN(R_2) \xrightarrow{\frac{W}{2} \times \frac{H}{2} \times 16} C(3 \times 3 \times 5); \quad K := BN([R_2, B_2]) \xrightarrow{\frac{W}{2} \times \frac{H}{2} \times 21} Pool_{avg}(2 \times 2);$ $K \xrightarrow{\frac{W}{4} \times \frac{H}{4} \times 21} FC(300) \xrightarrow{300} FC(W \cdot H)$. |
| **E2E** [16] | $R_2, B_2$ and $K$ like in FC-CNN; $\quad T := K \xrightarrow{\frac{W}{4} \times \frac{H}{4} \times 21} C^T(3 \times 3 \times 30);$ $[R_2, T] \xrightarrow{\frac{W}{2} \times \frac{H}{2} \times 46} C^T(6 \times 6 \times 30) \xrightarrow{W \times H \times 30} C(1 \times 1 \times 1)$. |
| | For every model defined below: $Z := C(5 \times 5 \times 28) \xrightarrow{W \times H \times 28} C(4 \times 4 \times 30) \xrightarrow{W \times H \times 30} C(3 \times 3 \times 30) \xrightarrow{W \times H \times 30} C(1 \times 1 \times 1)$. |
| **CoordConv** [5] | $[I, Coords(X, Y)] \xrightarrow{W \times H \times (T+2)} Z$. |
| **LI CNN** | $[I, LI(2)] \xrightarrow{W \times H \times (T+2)} Z$. |
| **LW CNN** | $[I, LW(2) \otimes_{(1,1)} I] \xrightarrow{W \times H \times (T+2)} Z$. |
| **LI + LW CNN** | $[I, LI(2), LW(2) \otimes_{(1,1)} I] \xrightarrow{W \times H \times (T+4)} Z$. |
| **Persistent LI + LW CNN** | $P := [LI(2), LW(2) \otimes_{(1,1)} I]; \quad [I, P] \xrightarrow{W \times H \times T} [C(5 \times 5 \times 30), P]$ $\xrightarrow{W \times H \times 34} [C(4 \times 4 \times 30), P] \xrightarrow{W \times H \times 34} [C(3 \times 3 \times 30), P] \xrightarrow{W \times H \times 34} C(1 \times 1 \times 1)$. |
| **LI + LW – I CNN** | $[LI(2), LW(2) \otimes_{(1,1)} I] \xrightarrow{W \times H \times 4} Z$. |
| **LW222 CNN** | $[I, LW(2) \otimes_{(2,2)} I] \xrightarrow{W \times H \times (T+2)} Z$. |
| **LW111 CNN** | $[LW \odot I] \xrightarrow{W \times H \times T} Z$. |

Here $[\cdot, \cdot, ..., \cdot]$ denotes concatenation; $C(w \times h \times n)$ denotes a 2D convolutional layer having $n$ kernels, each of size $w \times h$; $F(\cdot)$ is a reshaping function, the output shape can be inferred from the outgoing arrow; $\sigma(\cdot)$ denotes a sigmoid activation function; $FC(x)$ denotes a fully-connected layer with $x$ neurons; $Pool_{max}(w \times h)$ denotes a 2D max-pooling having pool-size of $w \times h$; similarly, $Pool_{avg}(w \times h)$ denotes average pooling; $ReLU(\cdot)$ denotes a rectified linear unit activation function; $BN(\cdot)$ denotes batch-normalization [27] followed by a ReLU activation; $LSTM(x)$ denotes an LSTM layer with $x$ neurons; $C^T(w \times h \times n)$ denotes a 2D-transposed convolutional layer with $n$ filters, each of size $w \times h$; $ConvLSTM(x)$ denotes a ConvLSTM layer [7] with $x$ neurons.

The "persistent" model variation introduced in Table 1 appends the initial LI and/or LW combination to the input of each layer. This enables the model to localize in all of the convolutional layers, rather than only in the first one. In this example, the same LIs/LWs of the model inputs are incorporated in all the layers (this is possible because all the layers have inputs of the same size), but unique settings and localizations of each layer's inputs could also be used.

Every trainable architecture, unless specified otherwise, has been trained for 100 epochs, with the objective to minimize mean squared error (MSE) using Adam [28] optimizer with standard parameter values: $l = 10^{-3}$, $\beta_1 = 0.9$, $\beta_2 = 0.999$, $\epsilon = 10^{-8}$. This gave enough training time for even the slowest-learning architectures to reach a validation plateau. Every architecture was implemented with Keras [29] framework using a TensorFlow [30] backend. To minimize the factor of luck, all the experiments were repeated five times with different random weight initializations, and the mean error values and their standard deviations (where they fit) are reported.

## 6. Datasets and Results

In this section, we test the different ways to localize CNNs empirically. To test our ideas, we began with simple models and a classical controllable synthetic "bouncing ball" benchmark dataset in Section 6.1, and then we moved on to three real-world geospatial wind speed prediction benchmark datasets in Sections 6.2, 6.3, and 6.4 respectively. On the real-world datasets we trained and tested state-of-the-art predictions models, as reviewed in Section 3.1, with and without our CNN localizations, specifically defined in Section 5. We present specifics of each dataset and the results of applying the models on them in corresponding subsections of the four sections.

### 6.1. Proof of Concept: A Bouncing Ball Task

We firstly tested our ideas in a controlled environment on the classical synthetic bouncing balls dataset [31] (publicly available at https://github.com/zhegan27/TSBN_code_NIPS2015). This dataset is interesting because it has both global and location-specific dynamics. While the ball movement "physics" is the same in all the places of the frame, it also changes at the boundaries which the balls bounce off. Thus, a model that can capture both the global and local dynamics could be beneficial in this case as opposed to the classical CNN which only models dynamics globally.

We tested our CNN localization ideas on this benchmark first before moving to the real-world wind speed prediction datasets. We assumed that the real-world data (among other tasks discussed in Section 2) also have both global and local dynamics. Thus, if our localizations can improve the prediction of the synthetic dataset, we expect them to do the same on the real-world ones.

The data consists of $30 \times 30$ monochrome image (frame) sequences of a white ball on a black background. The balls have a 7.2 pixel radius. They start at a random location inside of the frame moving in a random direction at a constant velocity. We generate 1600 samples for training, 500 samples for the first testing set and 500 samples for the second one. Each sample consisted of 25 consecutive ball movement frames as input and the task is to predict the frame five time steps after the last input frame as output. Figure 5 illustrates the dataset.

To have a better insight into the results, we introduced two types of testing data. In the first one all the ball trajectories were represented and in the second one only the predicted trajectories of balls bouncing off the walls and corners were selected, like in Figure 5.

We compared the following models:

- CNN: a two-layer CNN with filter sizes of $(20 \times 20)$ and $(10 \times 10)$ respectively. There are 22 filters in the first layer and 10 filters in the second. This network has 242,043 learnable parameters.
- LI CNN: a similar setup to CNN, except that it has 20 filters in the first layer but two $(30 \times 30)$ learnable inputs are added to learn location-based features. This network has 237,841 learnable parameters.

- CoordConv: a similar setup to LI CNN, except that it has 21 filters in the first layer and additional x and y coordinate inputs instead of the learnable ones [5]. This network has 247,842 learnable parameters.
- RandomConv: same setup to CoordConv, except that the two inputs were randomly initialized in $[-0.5, 0.5]$ range. The number of learnable parameters remained the same as in the CoordConv setup.

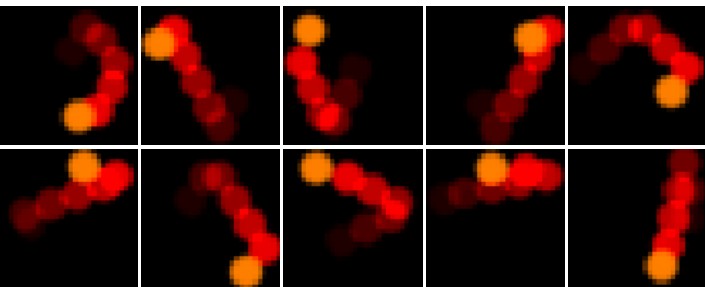

**Figure 5.** A few samples from the second testing set of the bouncing balls task. The sequence of frames was superimposed here into a single one. The red circles indicate input to the CNN (only every fourth frame is represented, more transparent red circles correspond to earlier frames in the sequence) and the orange one indicates the expected ground truth prediction. Note that every orange circle here is a result of hitting a wall or a corner.

We designed the networks in such a way that the comparison would be fair with respect to the number of learnable parameters. As a result, our proposed architecture had the smallest number of learnable parameters.

Each model was trained with five random initializations and each training session consisted of 100 epochs. We plotted the averaged validation and testing errors at each epoch in Figure 6. We can see that the CoordConv, the RandomConv, and the LI CNN consistently outperform CNN, which can be attributed to the models' ability to localize. Both CoordConv and RandomConv perform very similarly, as they allow the model to localize, but the local parameters are not learned to be task-specific. LI CNN always outperforms them both even though they have more trainable parameters. We also see that the bounced-off trajectories are harder to predict and the models capable of localization have a bigger relative advantage here compared to CNN, which confirms our hypothesis.

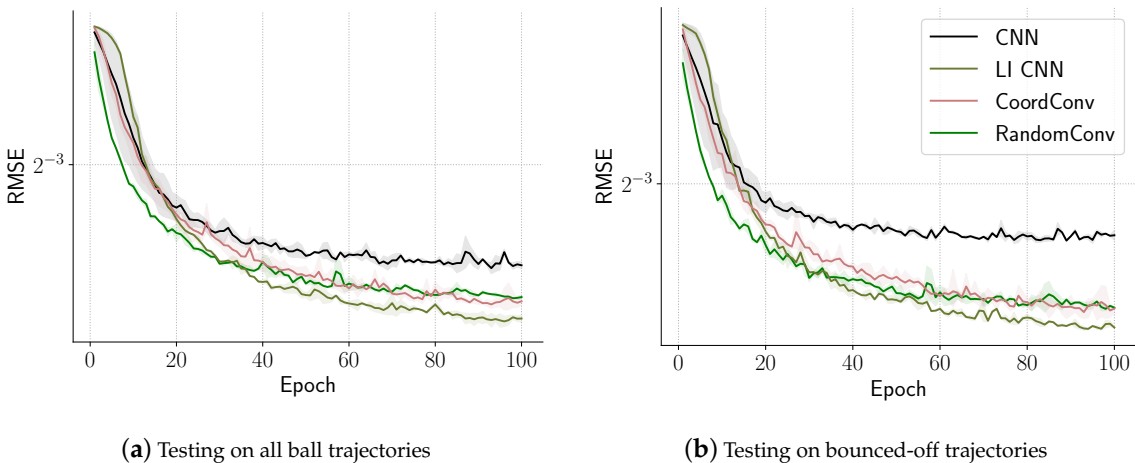

**(a)** Testing on all ball trajectories　　　　　　　　　**(b)** Testing on bounced-off trajectories

**Figure 6.** Averaged error on the bouncing ball task vs. epochs of training. Errors are plotted on a logarithmic scale. Shading indicates min/max error.

### 6.2. Case Study: Wind Integration National Dataset

We next evaluated the proposed methods on a real-world wind dataset from Indiana, United States. We used the Wind Integration National Dataset (WIND) [32] toolkit by The National Renewable Energy Laboratory (NREL), which contains the weather characteristics data from 126,000 stations covering years 2007–2013 with 5 min temporal sampling frequency. We selected a $(10 \times 10)$ sized rotated rectangular grid of wind turbines, where the left bottom point has GPS coordinates of $(85.215° \text{ W}, 40.4093° \text{ N})$ and the top right point has the coordinates of $(84.9684° \text{ W}, 40.2212° \text{ N})$ as depicted in Figure 7. We used only wind speed data of the first three months of 2012, which included 25,920 data frames. We chose this dataset, location, and time interval to make our results comparable to [13]. We confirm the match of the dataset both visually (see Figure 8) and by the characteristics reported in [13]: maximum wind speed at 27.228 m/s and minimum at 0.048 m/s.

We used the same setup of data preparations as in [13]: 60% of the data was used for training, 20% for validation, and the rest 20% for testing. Although not mentioned in [13], we also normalized the data to a $[0; 1]$ interval and de-normalize the prediction results before calculating the error. We carried out the experiments for forecast horizons of 5, 10, 15, 20, 30, and 60 min.

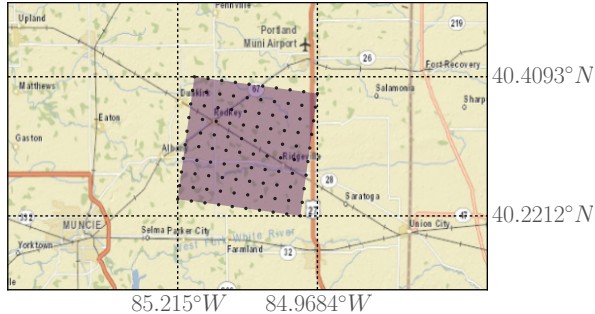

**Figure 7.** Locations used from the WIND dataset.

We can observe in Figure 8 a spatial pattern shifting through the grid, indicating that for short-term predictions simpler temporal models might suffice, given that the spatial model is powerful enough. However, for longer forecasting horizon the spatial model might not be as relevant. This is in part due to the fact that our grid covers a relatively small area.

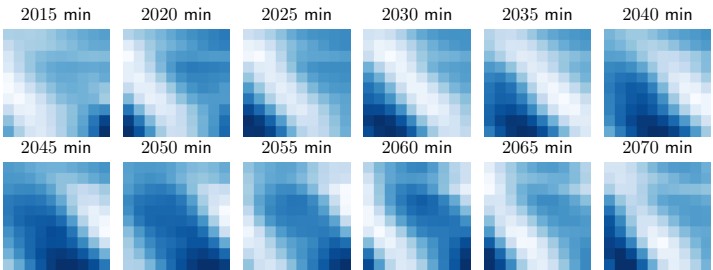

**Figure 8.** An hour of wind speeds used from the WIND dataset. The minutes are counted from the start of the dataset here.

We used both classical neural network methods, special architectures for such spatio-temporal prediction proposed in recent literature, and our variations of localized CNNs, as discussed in Sections 4 and 5. The RMSE results of different models are presented in Table 2.

Just as expected, we can see that for short-term forecast the spatial modeling is more important and temporal modeling becomes more important when forecast horizon increases. For 1-step (5-min) prediction, simple models without a dedicated temporal component are sufficient and in fact give the best results, while for longer prediction horizons, the LSTM often performed best, which has a strong temporal component and no spatial one at all.

For shorter forecast horizons, where spatial modeling is relevant, virtually every model which we augment with our learnable localized features gave a better performance compared to the vanilla versions.

Interestingly, most of the models that performed the best on the validation set, did not perform as well on the testing set. This is probably due to the fact that the models were trained on only one season of the year, winter, validated on the beginning of March, and tested on the rest of the month, sticking to the scheme in [13]. Dominating wind directions in Indiana depend on the time of the year (https://www.ncdc.noaa.gov/climatenormals/clim60/states/Clim_IN_01.pdf).

**Table 2.** RMSE results of different models and prediction horizons on the WIND dataset. Best testing results are indicated in bold.

| Model | 5 min | | 10 min | | 15 min | | 20 min | | 30 min | | 60 min | |
|---|---|---|---|---|---|---|---|---|---|---|---|---|
| | Valid | Test | Valid | Test | Valid | Test | Valid | Test | Valid | Test | Valid | Test |
| PR | 0.3929 | 0.3931 | 0.5848 | 0.5850 | 0.7232 | 0.7231 | 0.8333 | 0.8335 | 1.006 | 1.0079 | 1.3779 | 1.3785 |
| CNN | 0.2851 | 0.3429 | 0.4298 | 0.5009 | 0.5368 | 0.6085 | 0.6198 | 0.6991 | 0.7596 | 0.8685 | 1.0973 | 1.2228 |
| MLP | 0.3181 | 0.3617 | 0.4515 | 0.5038 | 0.5463 | 0.6054 | 0.6216 | 0.6911 | 0.7453 | 0.8460 | 1.0906 | **1.2169** |
| LSTM | 0.3039 | 0.3471 | 0.4366 | 0.5003 | 0.534 | **0.6020** | 0.6138 | 0.6899 | 0.7414 | 0.8444 | 1.0735 | 1.2322 |
| CoordConv [5] | 0.2864 | 0.3447 | 0.4323 | 0.5060 | 0.5402 | 0.6133 | 0.6228 | 0.7026 | 0.7668 | 0.8770 | 1.1007 | 1.2311 |
| ConvLSTM [7] | 0.3014 | 0.3730 | 0.4564 | 0.5438 | 0.5623 | 0.6706 | 0.6438 | 0.7668 | 0.7835 | 0.9259 | 1.137 | 1.3380 |
| PSTN [13] | 0.3373 | 0.3653 | 0.4572 | 0.5120 | 0.5504 | 0.6048 | 0.6245 | **0.6890** | 0.7503 | **0.8364** | 1.0804 | 1.2243 |
| PDCNN [15] | 0.4149 | 0.4358 | 0.5041 | 0.5412 | 0.5785 | 0.6290 | 0.6438 | 0.7064 | 0.7639 | 0.8604 | 1.0872 | 1.2509 |
| E2E [16] | 0.3636 | 0.4250 | 0.4809 | 0.5584 | 0.5714 | 0.6479 | 0.6472 | 0.7371 | 0.7769 | 0.8777 | 1.0973 | 1.2373 |
| FC-CNN [16] | 0.4131 | 0.4669 | 0.5164 | 0.5888 | 0.5861 | 0.6613 | 0.6635 | 0.7582 | 0.7288 | 0.8528 | 1.1074 | 1.2945 |
| LI CNN | 0.2825 | 0.3488 | 0.4289 | 0.5030 | 0.5354 | 0.6143 | 0.621 | 0.7062 | 0.7586 | 0.8656 | 1.0973 | 1.2385 |
| Persistent LI CNN | 0.2812 | 0.3400 | 0.4254 | 0.5014 | 0.5381 | 0.6135 | 0.6204 | 0.7026 | 0.7606 | 0.8616 | 1.0973 | 1.2311 |
| LW CNN | 0.2825 | 0.3438 | 0.4289 | 0.5018 | 0.5354 | 0.6082 | 0.6174 | 0.6980 | 0.7606 | 0.8669 | 1.094 | 1.2263 |
| LW111 CNN | 0.2825 | 0.3419 | 0.428 | 0.5021 | 0.5354 | 0.6098 | 0.6168 | 0.6990 | 0.7576 | 0.8621 | 1.094 | 1.2252 |
| LI + LW CNN | 0.2798 | 0.3378 | 0.4272 | 0.5032 | 0.534 | 0.6072 | 0.6186 | 0.6990 | 0.7591 | 0.8715 | 1.1007 | 1.2359 |
| Persistent LI + LW CNN | 0.2838 | 0.3445 | 0.4306 | 0.5037 | 0.5374 | 0.6091 | 0.6228 | 0.7077 | 0.7606 | 0.8665 | 1.0973 | 1.2303 |
| LI + LW222 CNN | 0.2798 | **0.3375** | 0.4272 | **0.4998** | 0.5326 | 0.6086 | 0.618 | 0.7036 | 0.7591 | 0.8651 | 1.0973 | 1.2350 |
| LI + LW – I CNN | 0.3014 | 0.3550 | 0.4433 | 0.5108 | 0.5483 | 0.6206 | 0.6304 | 0.7150 | 0.7721 | 0.8836 | 1.1107 | 1.2530 |
| Persistent LI PDCNN | 0.3995 | 0.4188 | 0.5019 | 0.5434 | 0.5727 | 0.6300 | 0.6438 | 0.7117 | 0.7576 | 0.8561 | 1.0804 | 1.2372 |
| LI + LW PDCNN | 0.4041 | 0.4305 | 0.4945 | 0.5377 | 0.5714 | 0.6215 | 0.6438 | 0.7042 | 0.7586 | 0.8462 | 1.0804 | 1.2397 |
| LI + LW PSTN | 0.3318 | 0.3594 | 0.4564 | 0.5070 | 0.5456 | 0.6055 | 0.624 | 0.6950 | 0.7498 | 0.8477 | 1.077 | 1.2296 |

*6.3. Case Study: Meteorological Terminal Aviation Routine Dataset*

We believe that localized CNNs could additionally be helpful when dealing with "non-orderly" embedded grid data. In this experiment, we use the dataset from the Meteorological Terminal Aviation Routine (METAR) weather reports of 57 stations in the East Coast including Massachusetts, Connecticut, New York, and New Hampshire (see [33] for details). This dataset consists of 6361 data points with hourly temporal sampling that are not embedded in a regular grid.

It is assumed that the data come in every 6 h, and we need to make a prediction for each hour until the new data come. We use 5700 samples for training, 300 for validation, and 361 for testing to make experiments compatible with [33]. We also used the same window size of $l = 12$. It is important to note that in [33] a different LSTM model was trained for every time step prediction (6 models in total); we instead make the six time step predictions at once as six different outputs of the same model to save processing time, although sliding window approach could also be used.

Only temporal data was available and no further information about each wind turbine location was included. To see if we could benefit from a spatial arrangement, we placed the locations on a regular rectangular grid. We first embedded these randomly-ordered 57 wind turbine stations into an $8 \times 8$ grid and padded the last row's final seven entries with zeros (they were not included in calculating errors). Since the positions of the stations were unknown to us, the embedding technique used in, e.g., [16] cannot be applied.

Placing locations randomly on a grid might not be very beneficial since CNNs make use of local correlations in the data. To get a better embedding we propose to order the grid based on mutual information of the location signals.

6.3.1. Mutual Information Based Grid Embedding

We interpret every temporal data of every turbine $i$ as a random variable $X_i$ and want to embed every turbine in a grid in such a way that the overall sum of correlation estimates in every neighborhood would be the highest. To determine the strength of similarity between the turbines, we rely on the mutual information (MI) [34] measurement, which is a popular technique quantifying the amount of information that two variables share together. For instance, a high level of MI between two variables means that the knowledge about one of the variables implies low uncertainty about the other variable. MI is defined as:

$$I(X, Y) = \sum_{y \in Y} \sum_{x \in X} p_{(X,Y)}(x, y) log \left( \frac{p_{(X,Y)}(x, y)}{p_X(x) p_Y(y)} \right), \tag{10}$$

where $p_{(X,Y)}(x, y)$ denotes the joint probability function of $X$ and $Y$, and $p_X(x)$, $p_Y(y)$ denote marginal probability function of both $X$ and $Y$. Note that variable independence implies that $I(X, Y) = 0$. Since neither marginal nor probability mass functions are known, we approximate these functions by creating discrete temporal time series of each wind turbine and calculate 1D histograms for both $p_X(x)$ and $p_Y(y)$ and 2D histogram for $p_{(X,Y)}$. Then we construct MI weights matrix $MI(i, j)$, which stores the MI between $i$ and $j$ wind turbines. After that, we use an evolutionary algorithm to embed the 57 values into a $8 \times 8$ grid. Concretely, 300 individuals were in the population and each member of the population represented some permutation of the 57 wind turbines and seven dummy elements. The probability of mutation was set to 0.2 and probability of crossover between two individuals was set to 0.3. The fitness $F(a)$ of an individual $a$ was evaluated by summing MI (10) between all the direct neighbors (including diagonal) on the $8 \times 8$ grid of that permutation.

We ran this algorithm with an objective to maximize $F(a)$ for 50,000 epochs. This experiment was carried out using DEAP library [35]. The original direct ordering of the grid had $F(a_{\text{direct}}) = 21.99$ and the optimized ordering $a_{\text{optimized}}$ found by the evolutionary algorithm had $F(a_{\text{optimized}}) = 35.96$, showing a significant increase in overall elements similarity among neighbors. Graphical representation of every location's contribution to $F(\cdot)$ is presented in Figure 9.

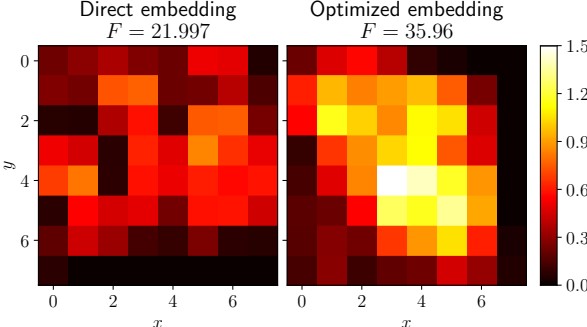

**Figure 9.** Each location's mutual information (MI) to its neighbors evaluated on the direct embedding (**left**) and the optimized embedding found with the evolutionary algorithm (**right**).

Even though such a grid embedding might help to handle non-grid-location data with CNNs in a reasonable way, it is still a rough approximation.

6.3.2. Results

To validate the proposed models, we construct every variation in such a way, that each model would have a roughly equal number of learnable parameters. We use all of the baselines described in Section 5 and both the optimized and the direct grid embeddings of the data. The results are presented in Table 3.

**Table 3.** Results on METAR data embedded on a regular grid, the direct and optimized embeddings. Models that disregard spatial relations are reported "–" results in the second, as they would be identical. Models that have no natural spatially-ordered output are marked with "*". ± denotes standard deviation. Best results are highlighted in bold.

| Model | Direct Embedding | | Optimized Embedding | |
|---|---|---|---|---|
| | **RMSE** | **MAE** | **RMSE** | **MAE** |
| DL-STF (All nodes) * [33] | 1.6200 | 1.1800 | – | – |
| PR | 1.791 ± 0.0000 | 1.238 ± 0.0000 | 1.791 ± 0.0000 | 1.238 ± 0.0000 |
| CNN | 1.527 ± 0.0059 | 1.140 ± 0.0057 | 1.509 ± 0.0084 | 1.119 ± 0.0066 |
| MLP * | 1.578 ± 0.0090 | 1.184 ± 0.0131 | – | – |
| LSTM * | 1.665 ± 0.0150 | 1.232 ± 0.0125 | – | – |
| CoordConv [5] | 1.524 ± 0.0064 | 1.134 ± 0.0066 | 1.519 ± 0.0135 | 1.124 ± 0.0085 |
| ConvLSTM [7] | 1.536 ± 0.0089 | 1.135 ± 0.0066 | 1.503 ± 0.0093 | 1.110 ± 0.0063 |
| PSTN * [13] | 1.724 ± 0.0228 | 1.293 ± 0.0149 | 1.716 ± 0.0113 | 1.271 ± 0.0152 |
| PDCNN * [15] | 1.696 ± 0.0173 | 1.277 ± 0.0151 | 1.696 ± 0.0120 | 1.265 ± 0.0100 |
| E2E [16] | 1.627 ± 0.0087 | 1.224 ± 0.0102 | 1.579 ± 0.0145 | 1.179 ± 0.0132 |
| FC-CNN * [16] | 1.676 ± 0.0210 | 1.255 ± 0.0182 | 1.676 ± 0.0145 | 1.251 ± 0.0076 |
| LI CNN | 1.518 ± 0.0088 | 1.133 ± 0.0074 | 1.505 ± 0.0094 | 1.118 ± 0.0048 |
| LW CNN | 1.516 ± 0.0062 | 1.132 ± 0.0044 | 1.508 ± 0.0091 | 1.120 ± 0.0044 |
| LW111 CNN | 1.511 ± 0.0038 | 1.129 ± 0.0053 | 1.506 ± 0.0055 | 1.122 ± 0.0043 |
| LI + LW CNN | 1.512 ± 0.0079 | 1.131 ± 0.0061 | 1.502 ± 0.0087 | 1.118 ± 0.0079 |
| Persistent LI + LW CNN | 1.507 ± 0.0072 | 1.124 ± 0.0062 | 1.501 ± 0.0077 | 1.111 ± 0.0050 |
| LI + LW222 CNN | 1.508 ± 0.0037 | 1.127 ± 0.0054 | 1.504 ± 0.0071 | 1.117 ± 0.0057 |
| Persistent LI + LW222 CNN | 1.507 ± 0.0044 | 1.126 ± 0.0066 | 1.499 ± 0.0072 | 1.116 ± 0.0057 |
| LI + LW − I CNN | 1.505 ± 0.0066 | 1.126 ± 0.0052 | 1.508 ± 0.0156 | 1.123 ± 0.0121 |
| **Persistent LI + LW − I CNN** | 1.496 ± 0.0046 | 1.116 ± 0.0047 | **1.492** ± 0.0095 | **1.106** ± 0.0083 |
| LI + LW222 − I CNN | 1.515 ± 0.0080 | 1.136 ± 0.0070 | 1.518 ± 0.0084 | 1.135 ± 0.0084 |
| Persistent LI + LW222 − I CNN | **1.492** ± 0.0058 | **1.113** ± 0.0058 | 1.496 ± 0.0061 | 1.109 ± 0.0082 |
| LI + LW PDCNN * | 1.677 ± 0.0054 | 1.254 ± 0.0086 | 1.675 ± 0.0138 | 1.251 ± 0.0115 |
| LI + LW PSTN * | 1.715 ± 0.0183 | 1.278 ± 0.0177 | 1.712 ± 0.0232 | 1.262 ± 0.0139 |

We can see in Table 3 that the Persistent LI + LW − I CNN model on the optimized embedding gave the best results by the MAE metric while performing as good as Persistent LI + LW222 − I model on the original embedding by the RMSE metric, and both showed the best overall results. The optimized embedding did decrease the testing error with every model that takes into account the spatial information and does not discard the input by RMSE error, while every model benefited from the optimized embedding by the MAE error metric. Models that discarded the input did not show an obvious benefit from the optimized embedding.

It is interesting that with the direct sub-optimal embedding the Persistent LI + LW222 − I CNN model which disregards the original input and is in principle capable of "swapping" adjacent inputs, performed best. This indicates that the model is learning a better input representation than the provided one, and including the latter (as in Persistent LI + LW222 CNN) gets even detrimental. This is not the case with the optimized embedding.

Every architecture that did not discard the spatial relations of the data in its temporal component (except for DL-STF, which had a different model for every prediction horizon), did perform significantly better than the other baselines marked with "*", even with the non-optimized embedding.

Comparing the performance of CNN, CoordConv, and our localized CNN models, we can see that in the case when the embedding is not perfect, learning the localized features is much more important, but just knowing the location (as in CoordConv) does not do much good, if any.

We see that, generally, the more location-specific features are learned on top of the CNN model (total number of learnable parameters was kept roughly the same), the better the performance gets on both embeddings. We could argue that the locally-learned features partially compensate for the defects of the embedding.

### 6.4. Case Study: Copernicus Dataset

Lastly, we test our localized CNN methods on a more spatially widely distributed wind speed dataset. For this we use wind speeds collected at 10 m above the surface level from "Climate data for the European energy sector from 1979 to 2016 derived from ERA-Interim" dataset (publicly available from https://cds.climate.copernicus.eu/cdsapp#!/dataset/sis-european-energy-sector?tab= overview). This dataset covers most of Europe with a resolution of $(0.5° \times 0.5°)$ with a temporal sampling frequency of 6 h. We selected a $10 \times 10$ region with the bounding rectangle of [52.75° N; 57.25° N], [6.75° W; 2.25° W], which is illustrated in Figure 10. Since the distances between the sites are much larger, so are the differences among the sites. We expect that while wind patterns have much in common at every location, in this scenario learning location-specific features in addition to convolutional neural networks becomes even more pertinent.

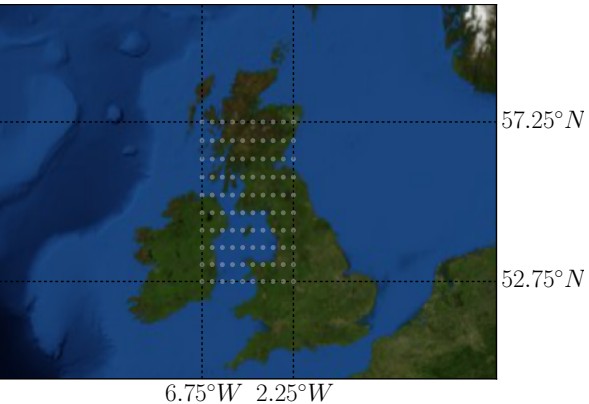

**Figure 10.** Locations used from the Copernicus dataset. Note the wide variety of the site locations, including both sea, high and low land.

More specifically, the dataset consists of 55,520 data frames, spanning a 37-year time window. We use 70% of the dataset for training, 10% for validation, and the final 20% for testing. The optimal window size was determined to be $l = 8$ by cross-validation. We test the models with the prediction horizon of $h = \{1, 2\}$, which corresponds to 6 and 12 h predictions respectively. The first 12 data frames are illustrated in Figure 11.

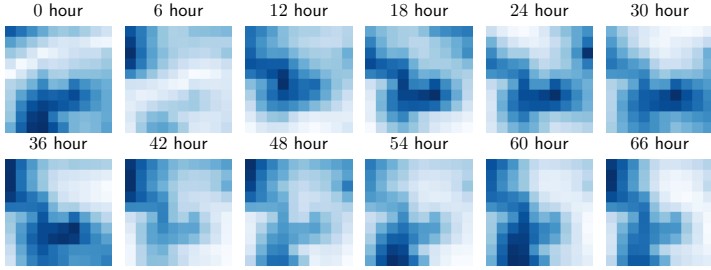

**Figure 11.** The first 72 h of wind speeds used from the Copernicus dataset.

We can immediately observe that the wind patterns are quite different above the sea and the land. This is confirmed by plotting global means and standard deviations of wind speeds across the training dataset in all locations, Figure 12. The winds tend to be the strongest and most varied at the most open area of the grid at the Atlantic Ocean (the top left corner).

Models used with this data were the same as those described in the previous section. The results are reported in Table 4. We also include a bigger version of PSTN [13] "PSTN bigger" where we increased the number of kernels in the first layer to 30 and ensured that the total number of learnable parameters is greater than in LI + LW PSTN.

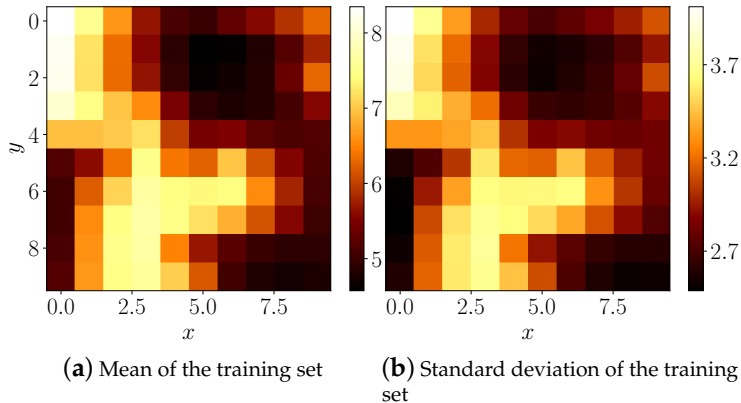

(**a**) Mean of the training set      (**b**) Standard deviation of the training set

**Figure 12.** Mean and standard deviation of the training set. Note the higher mean and standard deviation of wind speeds above the sea, compared to the land.

We can see in Table 4 that in general, the models that have a recurrent net LSTM component (LSTM, PSTN) perform best in each group. They are followed by the models that include both convolutional and dense layers (PDCNN, E2E, FC-CNN). Fully convolutional models (CNN variations, CoordConv) perform worst. This indicates that memory and/or longer spatial connections are required. This seems logical since this data a sparse both in space and in time.

Finally, we can once again see that all the models for which the convolutions were endowed with our learnable localization (LI, LW) performed better than the non-localized counterparts. Consequently, LI + LW PSTN showed the best overall results. Simply knowing the location (CoordConv vs. CNN) did not help in this case either.

**Table 4.** Results of different models and two prediction horizons on Copernicus dataset.Results in bold indicate lowest error in the corresponding column; $\pm$ denotes standard deviation.

| Model | 6 h | | 12 h | |
|---|---|---|---|---|
| | **RMSE** | **MAE** | **RMSE** | **MAE** |
| PR | $1.929 \pm 0.0000$ | $1.454 \pm 0.0000$ | $2.688 \pm 0.0000$ | $2.041 \pm 0.0000$ |
| MLP | $1.494 \pm 0.0013$ | $1.117 \pm 0.0026$ | $2.113 \pm 0.0047$ | $1.599 \pm 0.0041$ |
| LSTM | $1.438 \pm 0.0007$ | $1.068 \pm 0.0017$ | $2.075 \pm 0.0047$ | $1.564 \pm 0.0041$ |
| CNN | $1.491 \pm 0.0020$ | $1.114 \pm 0.0025$ | $2.126 \pm 0.0015$ | $1.609 \pm 0.0027$ |
| PDCNN [15] | $1.447 \pm 0.0023$ | $1.081 \pm 0.0015$ | $2.080 \pm 0.0063$ | $1.570 \pm 0.0071$ |
| FC-CNN [16] | $1.458 \pm 0.0032$ | $1.087 \pm 0.0038$ | $2.093 \pm 0.0055$ | $1.576 \pm 0.0054$ |
| E2E [16] | $1.466 \pm 0.0052$ | $1.092 \pm 0.0046$ | $2.103 \pm 0.0075$ | $1.585 \pm 0.0071$ |
| CoordConv [5] | $1.496 \pm 0.0039$ | $1.118 \pm 0.0027$ | $2.128 \pm 0.0020$ | $1.608 \pm 0.0002$ |
| PSTN original [13] | $1.433 \pm 0.0080$ | $1.064 \pm 0.0073$ | $2.084 \pm 0.0000$ | $1.568 \pm 0.0000$ |
| PSTN [13] bigger | $1.431 \pm 0.0023$ | $1.063 \pm 0.0025$ | $2.072 \pm 0.0039$ | $1.560 \pm 0.0004$ |
| LI CNN | $1.485 \pm 0.0021$ | $1.106 \pm 0.0019$ | $2.124 \pm 0.0014$ | $1.606 \pm 0.0026$ |
| LW CNN | $1.486 \pm 0.0022$ | $1.110 \pm 0.0005$ | $2.124 \pm 0.0021$ | $1.607 \pm 0.0024$ |
| LW111 CNN | $1.485 \pm 0.0022$ | $1.108 \pm 0.0030$ | $2.124 \pm 0.0025$ | $1.607 \pm 0.0035$ |
| LI + LW CNN | $1.485 \pm 0.0026$ | $1.108 \pm 0.0031$ | $2.124 \pm 0.0012$ | $1.607 \pm 0.0004$ |
| LI + LW − I CNN | $1.493 \pm 0.0043$ | $1.114 \pm 0.0038$ | $2.127 \pm 0.0016$ | $1.608 \pm 0.0029$ |
| LI + LW222 CNN | $1.480 \pm 0.0033$ | $1.105 \pm 0.0027$ | $2.122 \pm 0.0035$ | $1.604 \pm 0.0048$ |
| Persistent LI + LW CNN | $1.485 \pm 0.0020$ | $1.108 \pm 0.0011$ | $2.125 \pm 0.0016$ | $1.607 \pm 0.0011$ |
| LI + LW PDCNN | $1.442 \pm 0.0025$ | $1.075 \pm 0.0020$ | $2.071 \pm 0.0062$ | $1.563 \pm 0.0066$ |
| Persistent LI PDCNN CNN | $1.436 \pm 0.0055$ | $1.071 \pm 0.0034$ | $2.067 \pm 0.0028$ | $1.556 \pm 0.0022$ |
| **LI + LW PSTN** | **$1.420 \pm 0.0014$** | **$1.058 \pm 0.0015$** | **$2.061 \pm 0.0012$** | **$1.553 \pm 0.0046$** |

## 7. Conclusions and Future Directions

In this work, we have motivated localization of Convolutional Neural Networks (CNNs) to balance the learning of global and local features. Some previous examples of localizations include providing pixel coordinates to the convolution [5] or adding locally connected layers after all the

convolutional layers that extracted the features [4], as detailed in Section 3.2. We have proposed and investigated several new ways of doing this, including static additional random inputs, that work similarly to providing coordinates; and learnable localizations in the form of Learnable Inputs (LIs) and Local Weights (LWs). The localizations can be added to any spatial layers (all of which currently use convolutions) of architectures with minimal overhead. There are likely other ways to localize CNNs, not yet investigated. Investigating more alternatives to improve the models is one possible direction of future work.

We have demonstrated on a synthetic benchmark bouncing balls dataset and through numerous experiments in geospatial wind speed prediction (Section 6) that the learnable localizations increase the performance of virtually all current architectures, that have a spatial component, proposed in the literature for this type of tasks, often improving the state-of-the-art. We took special care in reproducing the results reported by other authors as close and making the experiments as unbiased as possible: taking statistics of several random weight initializations, making sure that the localized and the non-localized models have about the same number of trainable parameters, all models get plenty of training time. We share our code at https://github.com/oshapio/Localized-CNNs-for-Geospatial-Wind-Forecasting for future reproducibility and benefit to the research community.

We consider the consistency with which our proposed CNN localizations improve the predictions in different architectures on the tested application domains to be the strongest empirical result of this study. It empirically proves our conjecture that the total translation invariance of CNNs is a limitation in some application domains that can be overcome by learning local features, at the same time not losing the benefits of convolutions.

The consistently-improved wind speed predictions in this work would definitely lead to better decisions based on them. We are, however, not in a position to comment on the practical implications of these improvements in the absolute sense. The decisions made based on the predictions are beyond the scope of this work. This could be investigated in other more application/implementation-oriented contributions or practical projects.

Our work is more fundamental in this context and concentrates on improving the deep machine learning models for spatio-temporal prediction. In practical terms, however, substantial gains in prediction accuracy could also be made by using more detailed data. We were limited here in only using scalar wind speed data at every location, in some cases recorded at relatively sparse time intervals. Taking in more frequent data, wind direction, other meteorological and location-specific conditions into account could substantially further improve the prediction accuracies in absolute terms.

Geospatial applications, where the locations are fixed, are prime examples of tasks where pixel-level localizations are helpful. This includes weather forecasting, important for wind and solar energy production among other things. However, the localizations can be useful in other domains too, including statically-framed images or videos, as discussed in Section 2. When locations are not perfectly pixel-aligned, the localizations in the upper layers of the architecture may be more rational.

The localizations are also beneficial when the modeled grid in reality is not perfect, like we demonstrated in Section 6.3, as they can learn to account for the implicit imperfections. They could also account for other location-bound data deficiencies in other domains, like mismatched/uncalibrated sensors, dead pixels, static objects, watermarks, or overlays obtruding the image.

These considerations open many more possible directions for future research in many different CNN application domains.

**Supplementary Materials:** We share the source code of our proposed methods and experiments at a public repository https://github.com/oshapio/Localized-CNNs-for-Geospatial-Wind-Forecasting.

**Author Contributions:** Conceptualization, M.L., A.U., L.S.; methodology, A.U., M.L.; software, A.U., L.S.; validation, A.U., L.S.; formal analysis, A.U., M.L.; investigation, A.U., L.S.; writing—original draft preparation, A.U., M.L.; writing—review and editing, M.L., A.U.; visualization, A.U.; supervision, M.L.; funding acquisition, M.L. All authors have read and agreed to the published version of the manuscript.

**Funding:** This research was partially supported by the Research, Development and Innovation Fund of Kaunas University of Technology (grant No. PP-91K/19).

**Conflicts of Interest:** The authors declare no conflict of interest.

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
