# Peer review of "Localized Convolutional Neural Networks for Geospatial Wind Forecasting"

_energies, doi:10.3390/en13133440_

Round 1

Reviewer 1 Report

One of the best papers I have reviewed. It is very interesting and well prepared. Over whole text it is obvious that the authors are diligent.

Problem that is described in the paper is very important for energy sector as it concerns the forecasting of renewable energy sources. Proper and precise forecasting of wind speed is a first step to obtain reliable energy forecasts. In this very paper, authors have done a really wide research on usage of different neural networks for the purpose mentioned. Most of their work considered usage of CNN and making enhancements of CNN by different hybrid and complex systems to do precise forecasts. But they have also done, as it should be, a comparison to work of other authors and a comparison to other methods. And in this point they work is very valuable. Although they did not get results far better than other methods but they have proved that their approach is promising.

I would change first few sentences of the introduction. There are some format problems in conclusions section.   

Author Response

> Open Review
> (x) I would not like to sign my review report
> ( ) I would like to sign my review report
> English language and style
> ( ) Extensive editing of English language and style required
> ( ) Moderate English changes required
> (x) English language and style are fine/minor spell check required
> ( ) I don't feel qualified to judge about the English language and style
> Yes Can be improved Must be improved Not applicable
> Does the introduction provide sufficient background and include all relevant references?
> (x) ( ) ( ) ( )
> Is the research design appropriate?
> (x) ( ) ( ) ( )
> Are the methods adequately described?
> (x) ( ) ( ) ( )
> Are the results clearly presented?
> (x) ( ) ( ) ( )
> Are the conclusions supported by the results?
> (x) ( ) ( ) ( )
> Comments and Suggestions for Authors
>
> One of the best papers I have reviewed. It is very interesting and well prepared. Over whole text it is obvious that the authors are diligent.
>
> Problem that is described in the paper is very important for energy sector as it concerns the forecasting of renewable energy sources. Proper and precise forecasting of wind speed is a first step to obtain reliable energy forecasts. In this very paper, authors have done a really wide research on usage of different neural networks for the purpose mentioned. Most of their work considered usage of CNN and making enhancements of CNN by different hybrid and complex systems to do precise forecasts. But they have also done, as it should be, a comparison to work of other authors and a comparison to other methods. And in this point they work is very valuable.

Thank you!

> Although they did not get results far better than other methods but they have proved that their approach is promising.

We consider the consistency with which virtually every architecture using convolutional layers benefits from our localizations in this application domain under rigorous testing (the same number of trainable parameters, etc.) to be the most convincing result. We now highlight this in Section 7.

> I would change first few sentences of the introduction.

We have changed the wording, hopefully improving the paragraph, but we would like to keep the general message connecting this work to the global challenges of the world.

> There are some format problems in conclusions section.

The unnecessary line brakes are now fixed at the beginning of Section 7. They were a result of pagination and figure placement.

> Submission Date
> 19 May 2020
> Date of this review
> 26 May 2020 18:44:00

A general summary of the largest changes, based on the feedback of all three reviewers and/or our own initiative:

* The first sentences of the article are rephrased.
* Section 2: refined and expanded with several additional paragraphs to make the problem, the motivation, and the idea more clear.
* Section 3: we explained better what are the different relations of the previous approaches to this work, and how they are different.
* Section 4: minor additional clarifications.
* Sections 5 and 6: renamed and the relations among sections/subsections explained, motivated better. Also, better descriptions of the result tables.
* Section 7: renamed and significantly expanded to discuss the results and multiple future research directions.
* The resolution of Figure 7 has been reduced and thus the size of the whole document significantly.
* An additional spelling and grammar check of the entire article, several mistakes fixed.
* The linked source code is now publicly available.

Reviewer 2 Report

In this paper, the authors apply convolutional neural nets to predict wind speed data, and claims that similar methods can be applied to other spatial/temporal datasets. The claimed innovation is the introduction of “localized” model inputs and structures that are specific to certain spatial regions.

While it has certain academic merit in attempting to use a popular machine learning technique over a unique problem domain. The paper lacks clarity and depth. As a result, the reviewer recommends major revision prior to resubmitting for publication.

  • It is not clear to the reviewer that why a wind forecast on the time-scale of years could be useful for any practical application, what is the actual industrial/practical problem being solved as a result of the CNN modeling work? The authors should conduct a more thorough background research on the actual application area instead of just on the literature background on CNN itself. Otherwise, it is hard to convince the readers that the problem is actually useful and meaningful for solution.
  • It is not clear to the reviewer why local weights are required and is novel in terms of CNN architecture. The whole purpose of the CNN and other deep neural network architecture is to be free of structural biases (in terms of model form), and let the robust optimizer take care of both the meta-parameters and activation coefficients learning. Usually the issue of local weights can be resolved by simply adjusting the CNN architecture and size of the convolution kernels. A more theoretical justification of exploration on this topic would strongly strengthen the authors position.
  • Table 1 lists the candidate models being compared, and all of these are just CNN models with very different orders of magnitude of trainable parameters, how do we ensure that their training input requirements are satisfied? Comparing models of different architecture is not easy, sometimes the perceived improvements are simply due to faster convergence against the current training dataset of one model than another.
  • It is not clear to the reader why the bouncing ball problem is translatable to weather forecasting
  • Section 6.3.1 did not define F(a), the objective function of the optimization mathematically
  • Section 6.4 – is this the actual only industrial dataset modeled using the methodology? It is not clear how data collected at 6 hour intervals is useful for any practical purposes.
  • Table 4 – the RMSE / MAE of all the models cited are within the same order of magnitude, it is pretty abstract to the reviewer how these minute differences mean in terms of real-world applications – does a RMSE improvement of 30% translate to a better grid control strategy? Or some kind of wind turbine planning that can better harness the energy?
  • Section 7 is not a discussion but rather a summary
  • Is there no conclusion / future work section?

Author Response

> Open Review
> (x) I would not like to sign my review report
> ( ) I would like to sign my review report
> English language and style
> ( ) Extensive editing of English language and style required
> (x) Moderate English changes required
> ( ) English language and style are fine/minor spell check required
> ( ) I don't feel qualified to judge about the English language and style
> Yes Can be improved Must be improved Not applicable
> Does the introduction provide sufficient background and include all relevant references?
> ( ) ( ) (x) ( )
> Is the research design appropriate?
> ( ) ( ) ( ) (x)
> Are the methods adequately described?
> ( ) ( ) (x) ( )
> Are the results clearly presented?
> ( ) ( ) (x) ( )
> Are the conclusions supported by the results?
> ( ) ( ) (x) ( )
> Comments and Suggestions for Authors
>
> In this paper, the authors apply convolutional neural nets to predict wind speed data, and claims that similar methods can be applied to other spatial/temporal datasets. The claimed innovation is the introduction of “localized” model inputs and structures that are specific to certain spatial regions.
>
> While it has certain academic merit in attempting to use a popular machine learning technique over a unique problem domain. The paper lacks clarity and depth. As a result, the reviewer recommends major revision prior to resubmitting for publication.
>

Thank you for the (most) critical, but constructive remarks! They helped to improve the quality and clarity of the article.

> It is not clear to the reviewer that why a wind forecast on the time-scale of years could be useful for any practical application, what is the actual industrial/practical problem being solved as a result of the CNN modeling work? The authors should conduct a more thorough background research on the actual application area instead of just on the literature background on CNN itself. Otherwise, it is hard to convince the readers that the problem is actually useful and meaningful for solution.

This is probably a misunderstanding. Two out of three of our real-world datasets do span for multiple years, but this is just to have enough data to train the models (the third dataset also spans several years, but we took a shorter time span to be compatible with experiments by other authors). All our wind speed predictions (on all three datasets) are short-term: the prediction horizons ranging from 5 minutes (results in Table 2) to 12 hours (results in Table 4).

We are also limited by the wind speed datasets publicly available. Also, since our contribution improves the CNN-based models, we wanted to be comparable to the deep learning literature in this domain in both datasets and experiment setups as much as we could.

> It is not clear to the reviewer why local weights are required and is novel in terms of CNN architecture. The whole purpose of the CNN and other deep neural network architecture is to be free of structural biases (in terms of model form), and let the robust optimizer take care of both the meta-parameters and activation coefficients learning. Usually the issue of local weights can be resolved by simply adjusting the CNN architecture and size of the convolution kernels. A more theoretical justification of exploration on this topic would strongly strengthen the authors position.

The whole Section 2 is dedicated to explaining the fundamental limitation of CNNs and motivating our approaches. We have refined and expanded it with several additional paragraphs in the new revision, to hopefully make this more clear.

Namely:

The standard CNNs _do_ have structural biases. The main one that we are addressing is the complete translation invariance (or equivarience, "location agnosticism"): each location _must_ be treated the same, convolutional layers have no way to learn to treat them differently, no matter the hyper-parameter settings (number of filters, their sizes, etc.). This is based on what information is available to the units and can be stored in the form of learned parameters.

This limitation, coming from a different context, has also been previously explored by other authors in a contribution "An Intriguing Failing of Convolutional Neural Networks and the CoordConv Solution", R. Liu et al., 2018 [4]. (Their solution was to just add pixel coordinates as additional inputs. We explore this option among others, but show that we can do better).

Our approaches alleviate this structural bias of standard CNNs, allowing the models to learn more things than they can using standard CNNs.

> Table 1 lists the candidate models being compared, and all of these are just CNN models with very different orders of magnitude of trainable parameters, how do we ensure that their training input requirements are satisfied? Comparing models of different architecture is not easy, sometimes the perceived improvements are simply due to faster convergence against the current training dataset of one model than another.

There is a wide variety of models we test presented in Table 2. Some of them (MLP, LSTM) do not have convolutional layers at all.

Regarding the number of learnable parameters, we are following the principles below. We use the original hyper-parameters if possible in models that have been reported in the literature for the same tasks to reproduce and represent the approaches as closely as possible. We hand-tune the main hyper-parameters of classical approaches for best validation performance. Models that have fewer convolutions do to have more trainable parameters for the same number of inputs and layers, and are indeed more prone to overfitting -- one of the benefits of CNNs discussed in Section 2. When adding our proposed localizations to the CNN layers of the models we change the hyper-parameters so that the "localized" version has roughly the same number of hyper-parameters as the original non-localized version. This is typically done by reducing the number of filters in the first convolutional layer. We do this to ensure that the model structure gives the benefit and not the number of learnable parameters.

The training indeed converges at different speeds for different models. But the 100 epochs that we used were ample enough for even the slowest-learning models to reach a validation plateau. In addition, multiple runs with different random initializations have been performed to eliminate "luck". We added this explanation now at the end of Section 5.

> It is not clear to the reader why the bouncing ball problem is translatable to weather forecasting

It is not directly translatable. It is, however, a controllable spatio-temporal prediction task where location matters, like we assume wind prediction (among other tasks) to be.

We added the second paragraph in Section 6.1 to make this connection more explicit:

We test our CNN localization ideas on this benchmark first before moving to the real-world wind speed prediction datasets. We assume that the real-world data (among other tasks discussed in Section 2) also has both global and local dynamics. Thus, if our localizations can improve prediction of the synthetic dataset, we expect them to do the same on the real-world ones.

> Section 6.3.1 did not define F(a), the objective function of the optimization mathematically

As stated in the section, "The fitness F(a) of an individual a was evaluated by summing MI between all the direct neighbors (including diagonal) on the 8x8 grid of that permutation.", and MI is defined in Equation (10). We added a more explicit reference to the equation in the new revision.

> Section 6.4 – is this the actual only industrial dataset modeled using the methodology? It is not clear how data collected at 6 hour intervals is useful for any practical purposes.

We used three different real-world geo-spatial wind speed prediction datasets, what we could get access to, as presented in Sections 6.2, 6.3, and 6.4 respectively. All the three datasets have their own specifics, discussed in the corresponding sections, prediction horizons ranging from 5 minutes to 12 hours. We tried to maintain the same experimental conditions with the datasets that have been previously used for this task by other authors.

> Table 4 – the RMSE / MAE of all the models cited are within the same order of magnitude, it is pretty abstract to the reviewer how these minute differences mean in terms of real-world applications – does a RMSE improvement of 30% translate to a better grid control strategy? Or some kind of wind turbine planning that can better harness the energy?

Grid control strategies and wind turbine planning are beyond the scope of this article.
We demonstrate that adding our proposed localizations to the state-of-the-art geo-spatial wind speed prediction architectures consistently improves the predictions under rigorous testing (the same number of trainable parameters, multiple runs, etc.). We consider this to be our most convincing and valuable result. We only show how to improve the predictions. Better predictions would of course lead to better decisions, but the decisions are outside our domain of expertise. In our opinion, predicting the real-time pricing of energy would perhaps be the most immediate beneficiary.

We added these considerations to the Section 7.

> Section 7 is not a discussion but rather a summary
> Is there no conclusion / future work section?

We renamed Section 7 to "Conclusions and Future Directions" (as suggested also by another reviewer) and expanded it accordingly and significantly.

> Submission Date
> 19 May 2020
> Date of this review
> 02 Jun 2020 17:53:33
>

A general summary of the largest changes, based on the feedback of all three reviewers and/or our own initiative:

* The first sentences of the article are rephrased.
* Section 2: refined and expanded with several additional paragraphs to make the problem, the motivation, and the idea more clear.
* Section 3: we explained better what are the different relations of the previous approaches to this work, and how they are different.
* Section 4: minor additional clarifications.
* Sections 5 and 6: renamed and the relations among sections/subsections explained, motivated better. Also, better descriptions of the result tables.
* Section 7: renamed and significantly expanded to discuss the results and multiple future research directions.
* The resolution of Figure 7 has been reduced and thus the size of the whole document significantly.
* An additional spelling and grammar check of the entire article, several mistakes fixed.
* The linked source code is now publicly available.

Reviewer 3 Report

The paper presents an extensive analysis of novel approaches for wind forecasting with deep learning approaches focusing in the productivity improvement of renewable energy wind turbines. The topic of the paper is very interesting and relevant to the journal. The paper is well organized, and the methods quite well explained. However, there are some minor corrections that should be done before the acceptance of the paper. So my opinion is to be accepted after a minor revision.

Required corrections:

  • Section 2, CNN describing section too poor, an option is to remove it or to improve it.
    • for example, line 59, filters dimension and patch dimension are presented with the same letters (k x k), etc.
    • line 74 "The last feature is .." I think that you mean "feature map", ??
  • Section 2, lines 104 and 106, I would suggest to provide better description about the two bullets.
  • Section 3, I would suggest to close the section by providing a shot description of the advantages of the contribution of the paper over the presented related works. 
  • Table2, it is not clear the units of measurement of values.
  • In general it would be good idea to present also prediction accuracy, not only the error. 
  • I would suggest to improve the highlights of the proposed approaches in the last section (7), to rename it to "Conclusions and future directions" and also to highlight some important future directions and some key future challenges.
  • A general spelling and grammar check in the entire paper.

Overall, the paper is interesting and relevant to the journal, I would suggest after a minor revision based on the instructions of the reviewers the paper to be accepted.

Author Response

> Open Review
> (x) I would not like to sign my review report
> ( ) I would like to sign my review report
> English language and style
> ( ) Extensive editing of English language and style required
> ( ) Moderate English changes required
> (x) English language and style are fine/minor spell check required
> ( ) I don't feel qualified to judge about the English language and style
> Yes Can be improved Must be improved Not applicable
> Does the introduction provide sufficient background and include all relevant references?
> (x) ( ) ( ) ( )
> Is the research design appropriate?
> ( ) (x) ( ) ( )
> Are the methods adequately described?
> (x) ( ) ( ) ( )
> Are the results clearly presented?
> ( ) ( ) ( ) ( )
> Are the conclusions supported by the results?
> ( ) (x) ( ) ( )
> Comments and Suggestions for Authors
>
> The paper presents an extensive analysis of novel approaches for wind forecasting with deep learning approaches focusing in the productivity improvement of renewable energy wind turbines. The topic of the paper is very interesting and relevant to the journal. The paper is well organized, and the methods quite well explained. However, there are some minor corrections that should be done before the acceptance of the paper. So my opinion is to be accepted after a minor revision.

Thank you! The suggestions bellow really helped to improve the article.

> Required corrections:
>
> Section 2, CNN describing section too poor, an option is to remove it or to improve it.

We believe that CNNs are well known and established as a staple of deep learning at this stage. The goal of this section is not a definitive description or introduction to CNNs, but to introduce our view on CNNs giving rise to the localizations conceptually, as well as notation. We have improved this section with more elaboration.

> for example, line 59, filters dimension and patch dimension are presented with the same letters (k x k), etc.

Yes a filter (kernel) of size (k x k) is applied to the same size (k x k) patch of an image. It is hopefully more clear in the new revision. We have now updated and expanded Section 2 significantly.

> line 74 "The last feature is .." I think that you mean "feature map", ??

We meant the last feature of CNNs in the list just above. It is now rewritten for more clarity.

> Section 2, lines 104 and 106, I would suggest to provide better description about the two bullets.

We elaborated on this significantly. However, the whole Section 4 is to define them properly.

> Section 3, I would suggest to close the section by providing a shot description of the advantages of the contribution of the paper over the presented related works.

The works reviewed in Section 3 are related in different ways to our contribution. Thus, instead of a single place, we added/improved the case-by-case explanations throughout the Section 3 on how they relate to our contributions and how the latter are different. The rest of the paper is to argue and demonstrate how these differences lead to advantages.

> Table2, it is not clear the units of measurement of values.

These are RMSE errors, homogenized and clarified now.

> In general it would be good idea to present also prediction accuracy, not only the error.

Our main goal is to compare different methods using the same metrics. We believe that our result tables are probably already too big. In fact we have left quite some variations of models off this article for this reason, and also some standard deviations. Adding accuracies would make the tables twice as big.

> I would suggest to improve the highlights of the proposed approaches in the last section (7), to rename it to "Conclusions and future directions" and also to highlight some important future directions and some key future challenges.

As suggested, we renamed Section 7 to "Conclusions and Future Directions" and expanded it accordingly and significantly.

> A general spelling and grammar check in the entire paper.

We have went through the entire article and have fixed several spelling and grammar mistakes.

> Overall, the paper is interesting and relevant to the journal, I would suggest after a minor revision based on the instructions of the reviewers the paper to be accepted.
>
> Submission Date
> 19 May 2020
> Date of this review
> 01 Jun 2020 16:26:07
>

A general summary of the largest changes, based on the feedback of all three reviewers and/or our own initiative:

* The first sentences of the article are rephrased.
* Section 2: refined and expanded with several additional paragraphs to make the problem, the motivation, and the idea more clear.
* Section 3: we explained better what are the different relations of the previous approaches to this work, and how they are different.
* Section 4: minor additional clarifications.
* Sections 5 and 6: renamed and the relations among sections/subsections explained, motivated better. Also, better descriptions of the result tables.
* Section 7: renamed and significantly expanded to discuss the results and multiple future research directions.
* The resolution of Figure 7 has been reduced and thus the size of the whole document significantly.
* An additional spelling and grammar check of the entire article, several mistakes fixed.
* The linked source code is now publicly available.

Round 2

Reviewer 2 Report

The authors addressed most of the concerns I had in the original review. I still have questions with regard to the actual practical use-case of this model; but nevertheless, the authors did present a good comparison of their approach against other state of art methods. I think that in itself has merit for publication.